# Using Trophy Hunting to Save Wildlife Foraging Resources: A Case Study from Moyowosi-Kigosi Game Reserves, Tanzania

Nyangabo V. Musika [1,*], James V. Wakibara [2], Patrick A. Ndakidemi [1] and Anna C. Treydte [1,3,4]

1   Department of Sustainable Agriculture, Biodiversity and Ecosystem Management, School of Life Sciences and Bio-Engineering, The Nelson Mandela African Institution of Science and Technology, P.O. Box 447, Arusha 23301, Tanzania; patrick.ndakidemi@nm-aist.ac.tz (P.A.N.); anna.treydte@nm-aist.ac.tz (A.C.T.)
2   College of African Wildlife Management—Mweka, P.O. Box 3031, Moshi 25215, Tanzania; james.wakibara2@mwekawildlife.ac.tz
3   Department of Physical Geography, Stockholm University, 10691 Stockholm, Sweden
4   Ecology of Tropical Agricultural Systems, Hans-Ruthenberg Institute, Hohenheim University, 70599 Stuttgart, Germany
*   Correspondence: nyangabomusika@yahoo.co.uk

**Abstract:** Globally, the role of trophy hunting in wildlife conservation has been a topic of much debate. While various studies have focused on the financial contribution of trophy hunting towards wildlife conservation, little is known about whether hunting activities can protect wildlife forage resources. We examined the effect of illegal livestock grazing on wildlife habitat in operational and non-operational wildlife hunting blocks in Moyowosi-Kigosi Game Reserves (MKGR), Tanzania. We assessed whether the physical presence of hunting activities lowered illegal grazing and, thus, led to higher vegetation quality. We compared 324 samples of above-ground biomass (AGB) and grass cover between control (0.0007 cattle ha$^{-1}$), moderately (0.02 cattle ha$^{-1}$), and intensively (0.05 to 0.1 cattle ha$^{-1}$) grazed hunting blocks. Likewise, we assessed soil infiltration, soil penetration, soil organic carbon (SOC), and soil Nitrogen, Phosphorus, and Potassium (N-P-K) across grazing intensity. Illegal grazing decreased AGB by 55%, grass cover by 36%, soil penetration by 46%, and infiltration rate by 63% compared to the control blocks. Illegal grazing further lowered SOC by 28% ($F_{2,33} = 8$, $p < 0.002$) but increased soil N by 50% ($F_{2,33} = 32.2$, $p < 0.001$) and soil K by 56% ($H(2) = 23.9$, $p < 0.001$), while soil P remained stable. We further examined if Hunting Company (HC) complements anti-poaching efforts in the Game Reserves (GR). We found that HC contributes an average of 347 worker-days$^{-1}$ for patrol efforts, which is 49% more than the patrol efforts conducted by the GR. However, patrol success is higher for GR than HC ($F_{1,21} = 116$, $p < 0.001$), due to constant surveillance by HC, illegal herders avoided invading their hunting blocks. We conclude that illegal grazing severely reduced vegetation and soil quality in MKGR. We further claim that trophy hunting contributes directly to wildlife habitat preservation by deploying constant surveillance and preventing illegal grazing. We propose maintaining trophy hunting as an essential ecological tool in wildlife conservation.

**Keywords:** cattle grazing; patrol efforts; anti-poaching; grass biomass; soil compaction

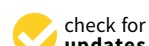



## 1. Introduction

Trophy hunting is a type of wildlife tourism whereby tourists pay to hunt and get trophies from wild animals such as hooves, tusks, skins, antlers, horns, teeth, claws, bones, feathers, or hair [1,2]. The phrase is used interchangeably with 'safari hunting', 'sport hunting', 'hunting tourism', 'recreational hunting', or 'tourist hunting' [3,4]. Trophy hunting is practiced worldwide in North America, South America, Australia, Europe, Canada, Asia, and 50% of African countries [5]. Currently, the trophy hunting industry is in a difficult position globally, receiving criticism concerning its ethical practicality and efficacy in wildlife conservation and community development [6].

Despite the criticism, trophy hunting is still extensively used to address social, ecological, and economic needs in developed and developing countries. When trophy hunting is well regulated and supervised efficiently, it generates financial incentives that encourage governments, private and communal landowners to conserve wildlife [5,7]. For instance, in North America, trophy hunters have contributed a substantial amount of money that enhanced wildlife research, law enforcement, and restoration of bighorn sheep (*Ovis canadensis*) [5,8]. In Pakistan, trophy hunting generated income, employed poachers, and the formerly small populations of the Afghan urial (*Ovis orientalis*) were conserved [9]. In Namibia, revenue generated from the hunting business supported the white rhino's recovery and protection (*Ceratotherium simum*) [7]. While trophy hunting's socio-economic benefits are reasonably well studied, research on its direct contribution to altering wildlife habitats and reducing illegal wildlife activities remains scarce.

In Tanzania, trophy hunting is considered one of the most economically viable forms of wildlife use that conserves wildlife within and outside Protected Areas (PAs) [10,11]. Forty per cent of Tanzania's land is legally protected [12]. Trophy hunting is practiced in 305,000 km$^2$, representing about 80% of Tanzania's total protected land [7]. Apart from generating direct income [10], it also gives wildlife an economic value so that wildlife habitats compete with other land uses in a self-funding way [13]. Despite its importance, there is limited information about how trophy hunting activities can directly contribute to wildlife conservation and their habitats [5]. Further, in Tanzania, the last decade has seen an escalation of incidences of illegal livestock grazing in PAs, which may affect wild mammalian herbivore species and their habitat by reducing vegetation biomass [14] and cover [15]. Grass cover protects the soil and improves soil infiltration capacity as their roots enhance soil aeration, and their decomposition increases soil organic matter [15,16]. Wild mammalian grazers often compete with domestic grazers for foraging resources, as was shown for African buffalo (*Syncerus caffer*) and cattle (*Bos primigenius* f. taurus) in East Africa [17]. Hence, livestock, in particular cattle, in high densities might strongly reduce foraging resources, deteriorate habitat quality for wildlife [18], decreases the number of wildlife, affects the ratio of ungulate to carnivores, minimizes wildlife activities during the day, and increases human activities in that PA [19]. Our research investigated whether trophy hunting is a deterrent against the illegal livestock intrusions in PAs [20], thereby keeping the impacts of overgrazing low. We tested the effect of hunting activities on wildlife habitat quality, focusing on habitat quality for African buffalo. This species is preferred by trophy hunters [21] and competes with livestock for forages and spaces [17]. We compared above-ground biomass (AGB), grass cover, soil penetration, soil infiltration, and selected soil physical-chemical properties [22] across a gradient of livestock grazing pressure, i.e., intensively, moderately, and control (not illegally) grazed hunting blocks in the Moyowosi-Kigosi Game Reserve (MKGR) of western Tanzania.

We hypothesized that, regardless of the amount of rainfall, areas of little or no hunting activity will show the highest livestock intrusion effects, with the lowest AGB, grass cover, and reduced soil infiltration and penetration capacity. We also expected that illegal grazing in the Game Reserve with the associated low AGB, grass cover, soil infiltration rate, and soil penetration capacity would affect essential soil physical-chemical properties such as SOC and N-P-K. Furthermore, we expected that areas with healthy grass biomass and cover would show high soil infiltration rates and penetration capacity. We further expected that the hunting company would complement the anti-poaching efforts of the Game Reserve authorities, particularly during the dry season, i.e., when hunting activities take place.

## 2. Material and Methods

### 2.1. Study Area

We conducted our study in the Moyowosi-Kigosi Game Reserve (MKGR), western Tanzania, located between 3°15′ and 5°00′ S, and 30°30′ and 32°00′ E (Figure 1). The MKGR was gazetted in 1981 and 1983 and covered 6000 km$^2$ and 7000 km$^2$, respectively [23]. Together with Ugalla Game Reserve, this area covers 75% of the Malagarasi-Muyovozi

Ramsar site, which comprises the Malagarasi River basin, which forms 30% of Lake Tanganyika's total catchment area, the world's second deepest freshwater lake [24], making various regions of the Reserve wet and impassable throughout the year.

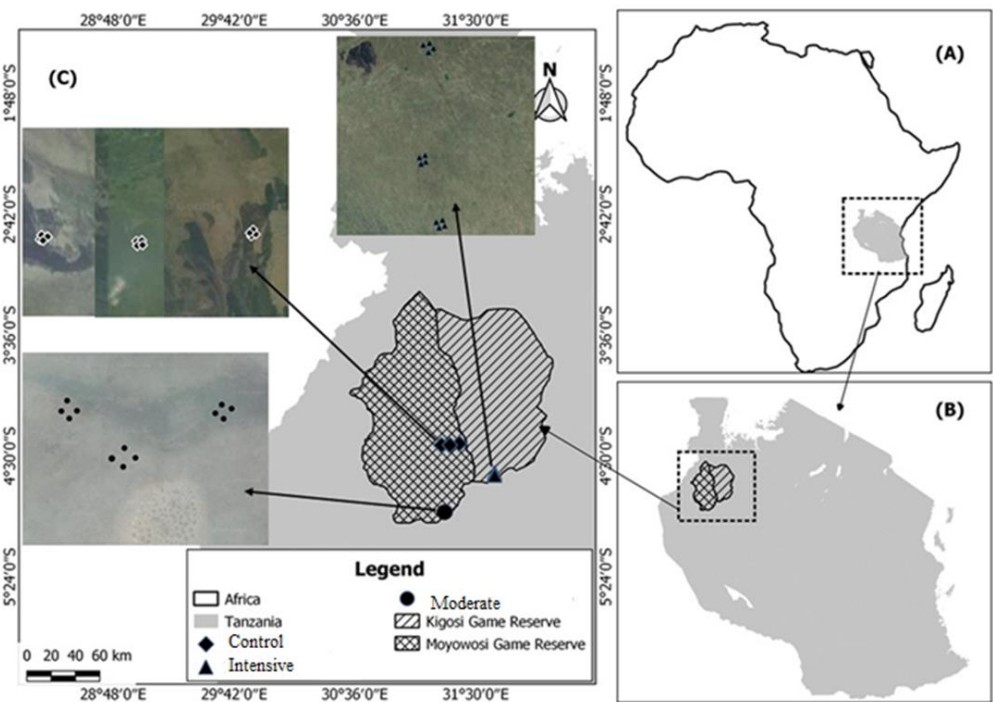

**Figure 1.** (**A**) Location of Tanzania within Africa, (**B**) Location of study site within Tanzania, (**C**) A map of Moyowosi-Kigosi Game Reserve showing the study areas with wildlife hunting blocks of different grazing intensities, i.e., the diamond shape (♦) indicates a hunting block with little or no livestock grazing = control, the round shape (●) indicates a hunting block which is moderately grazed = moderate, and the triangle shape (▲) indicates a hunting block that is intensively grazed = intensive. The hunting blocks were laid out and visited in the sampling periods of July 2019, October 2019, and July 2020.

The MKGR is rich in biodiversity and harbors wetland-dependent animals like the African slender-snouted crocodile, *Crocodylus cataphractus*, which is critically endangered [25] and bird species such as the shoebill (*Balaeniceps rex*) and the wattled crane (*Bugeranus caruncuta*). The mammalian species include African elephant (*Loxodonta africana*), lion (*Panthera leo*), giraffe (*Giraffa camelopardalis*) and African buffalo, all of which are vulnerable [25] while the buffalo is near threatened [25]. Miombo woodland is the primary vegetation dominated by the near threatened African black wood (*Dalbergia melanoxylon*) and blood wood (*Pterocarpus angolensis*) [25]. Managing MKGR is particularly challenging due to its extensive wetland nature, remoteness, and inaccessibility [2]. Its proximity to politically unstable countries adds to the pressure (mainly from illegal immigrants) of indiscriminate grazing and illegal natural resource harvesting [12]. As a result, illegal livestock grazing is high in MKGR [26]. Four of MKGR's nine hunting blocks have progressively lost their economic viability and surrendered to the Government [26]. Illegal grazing has intensified over 30 years in MKGR [18], and the aerial census of 2014 showed that livestock numbers inside MKGR were more than 50,000 heads while buffaloes numbers were only 2869 heads [27]. The blocks that hunters had vacated have not only remained particularly vulnerable to poaching and further livestock incursions but have also compromised the economic gains for the local communities and the Tanzanian economy [26].

### 2.2. Data Collection

Our research combined experimental fieldwork, laboratory work, and official records from Game Reserves' authorities in quantifying the impacts of illegal grazing on wildlife habitat and the roles played by the hunting company to counteract the effect. We also gathered questionnaires of 41 key informants, i.e., hunters and PAs managers, to supplement our data.

We obtained data on the encountered number of illegal grazing events from the MK-GRs' official records to determine grazing intensity and categorize hunting blocks into three different grazing intensities. Fully occupied hunting blocks that received maximum surveillance from hunters and showed little or no livestock grazing activities were taken as 'control' (Table 1). Those blocks with partial occupation and hunting activities were termed 'moderately grazed' (moderate) and blocks fully vacated by the hunting companies were classified as 'intensively grazed' (intensive). The control block covered 177,400 ha, the moderate 301,300 ha, and the intensive block 200,800 ha. We used the annual number of livestock encountered by rangers for the 2015–2019 years to calculate livestock densities for each hunting block category. We applied the thresholds of <0.001 cattle ha$^{-1}$, 0.02 cattle ha$^{-1}$, and 0.05–0.10 cattle ha$^{-1}$ to represent the non-grazed (control), moderately grazed, and intensively grazed blocks, respectively (see also [28]), Table 1.

**Table 1.** Livestock numbers (Number) and calculated livestock density (cattle ha$^{-1}$) per wildlife hunting block for five consecutive years to determine the intensity of grazing pressure for each block. Intensive intensively grazed, moderate moderately grazed by livestock, control no livestock grazing. The control block size was 177,400 ha, that of the moderate block was 301,300 ha, and the intensive block size was 200,800 ha.

| Years | Control | | Moderate | | Intensive | |
|---|---|---|---|---|---|---|
| | Number | Density | Number | Density | Number | Density |
| 2015 | 100 | 0.0006 | 6087 | 0.02 | 13,117 | 0.07 |
| 2016 | 120 | 0.0007 | 6980 | 0.02 | 13,397 | 0.07 |
| 2017 | 90 | 0.0005 | 5830 | 0.02 | 19,397 | 0.10 |
| 2018 | 0 | 0.0000 | 5767 | 0.02 | 12,716 | 0.06 |
| 2019 | 0 | 0.0000 | 5127 | 0.02 | 10,731 | 0.05 |

To assess the impact of illegal grazing on wildlife habitat quality, we evaluated the above-ground grass biomass (AGB), grass cover, and soil compaction level [22], as well as soil chemical properties across the three different hunting block categories. We sampled a total of nine plots of 100 m × 100 m in each hunting block category to assess the habitat quality of wildlife, particularly large grazer species such as the buffalo, using grass biomass, grass cover, and soil compaction [22] (Figure 1). Additionally, soil samples were taken to the lab for analysis to determine N, P, and K nutrient levels. We collected soil samples over three sampling periods for two consecutive calendar years (July 2019, October 2019, and July 2020). For AGB estimation, we randomly placed twelve 1 m × 1 m quadrats within each of the nine sample plots, thus yielding a total of 108 quadrats from control, moderate, and intensive blocks, i.e., 324 samples over the three sampling periods. In every quadrat, we harvested fresh AGB to ground level [29] and weighed it in the field using an Escali Primo model weighing balance [30]. We then oven-dried the fresh biomass at 60 °C for 60–72 h to constant weight [31].

We visually estimated the total herbaceous layer cover (as %) in every 1 m$^2$ quadrat before harvesting the fresh biomass. Additionally, for each quadrat, we identified the three most dominant herbaceous species with the help of a field guide [32] and local botanists. We measured soil penetration and infiltration across four measurement locations within each sampled quadrat [22,33].

To examine if there are variations on anti-poaching between GR and HC, we collected data for patrol efforts and patrol success for GR and HC for 2012–2016. Patrol efforts are

the amount of anti-poaching patrol conducted by one ranger per day or months while patrol success is the number of illegal people arrested or illegal activities encountered when rangers are conducting anti-poaching patrols [34,35]. Patrol efforts were obtained by multiplying the number of rangers involved in the patrol and the number of days they spend conducting the patrol patrols [35–37]. The most common unit used for patrol efforts is man-days$^{-1}$, this unit is changed to worker-days$^{-1}$ to avoid gender biases [18]. We also administered a structured questionnaire to 41 key informants to complement our field data. These informants included Wildlife Rangers and Managers from MKGR, Professional Hunters, District Wildlife Officers, and District Livestock Officers. We asked about the number of livestock encountered by rangers, seasonal challenges of illegal grazing, and strategies to minimize illicit activities in the Game Reserve.

### 2.3. Data Analysis

We transformed data for AGB, soil penetration capacity, and soil infiltration rate to meet normality and parametric analysis conditions. The SOC was analyzed using the Walkly and Black method [38], while we used Semi Micro Kjeldahl method for total nitrogen (TN) and Bray and Kutz No. 1 for available phosphorus (P) [39,40]. We used repeated measures ANOVA to test variation of AGB, grass cover, soil penetration capacity, and soil infiltration rate across different grazing categories and sampling periods. We also used one-way ANOVA to determine soil nutrient variation across different grazing categories [41].

We deployed Tukey's HSD post hoc tests to identify AGB differences, grass cover, soil penetration capacity, soil infiltration rate, soil nutrients across grazing treatments, and the sampling period. We conducted non-parametric analyses using the Kruskal–Wallis test followed by the Games–Howell post hoc test to determine the variation of soil Phosphorus and soil Potassium across different grazing intensities. We performed a correlation analysis between grass biomass, grass cover, soil penetration capacity, and soil infiltration rate to examine the influence of grass biomass and grass cover on soil compaction. To assess the variation of patrol efforts and patrol success between GR and HC during dry and wet seasons, we log-transformed data for patrol efforts and patrol success to meet normality conditions. We further performed one-way ANOVA to observe significant differences in the patrol efforts and patrol success between HC and GR. Tukey post hoc test was used to identify the season with high patrol efforts and patrol success for both HC and GR. Content analysis and descriptive statistics were used for interview data, and we performed Statistical analysis using Jamovi version 1.2 [42] and R version 4.0.3 [43].

## 3. Results

### 3.1. Illegal Grazing Lowers Grass Biomass and Grass Cover in Non-Operational Wildlife Hunting Blocks

We found that the grass AGB and cover differed significantly across grazing categories ($F_{2,315}$ = 503.0, $p < 0.001$ and $F_{2,315}$ = 157.4, $p < 0.001$, respectively; Figure 2) and across sampling period (biomass: $F_{2,315}$ =11.8, $p < 0.001$; cover: $F_{2,315}$ = 3.0, $p < 0.001$). The mean AGB in moderate and intensive blocks was 33% and 55% lower, respectively, than the control across all sampling periods (Figure 2A). Equally, the mean grass cover in moderate and intensive blocks was 14% and 36% lower, respectively, than that of control blocks in all sampling periods (Figure 2B). There was an interaction between grazing intensity and sampling periods for AGB ($F_{4,315}$ = 12.0, $p < 0.001$), but not for grass cover ($F_{4,315}$ = 1.2, $p$ = 0.20). Tukey's post hoc test indicated that the mean AGB in October 2019 sampling period was 9% slightly higher in the control hunting blocks than in July 2019 and July 2020 periods. When subjected to grazing, the mean AGB in moderate and intensive blocks of October 2019 was 24% lower than the other sampling periods, highlighting that grazing overruled the effect that rainfall might have (Figure 2A).

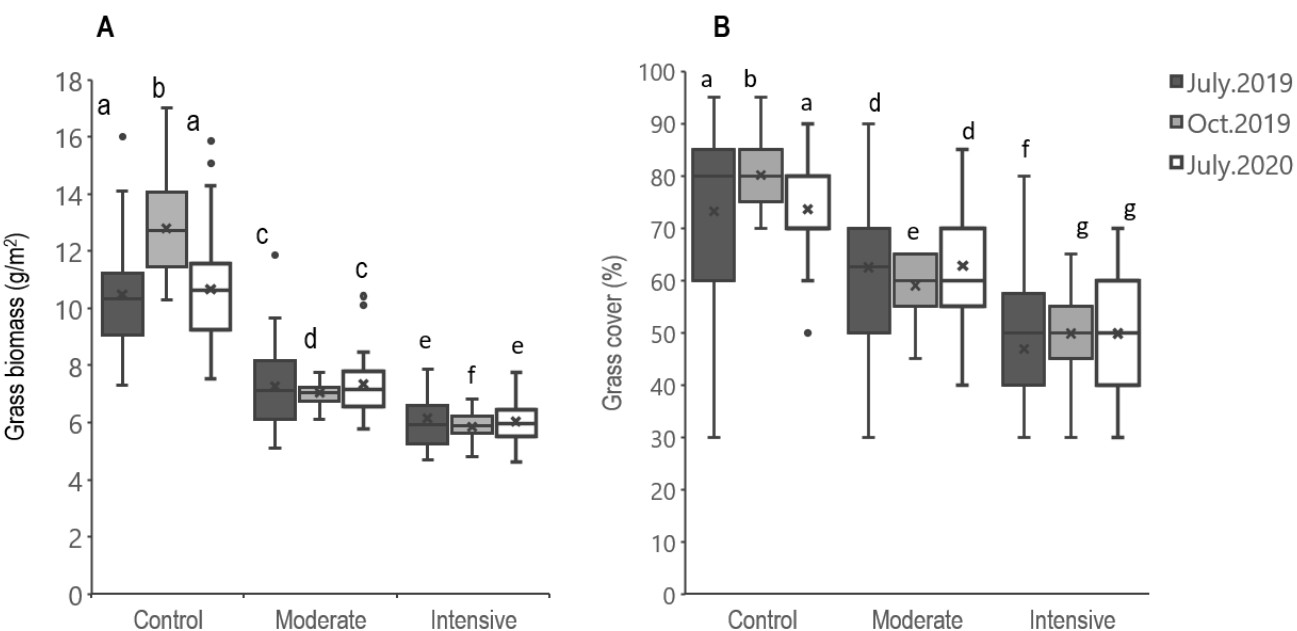

**Figure 2.** Boxplots of above-ground grass biomass (**A**) and cover (**B**) across different grazing intensities and sampling periods in an operational wildlife hunting block with little or no livestock grazing (Control), a non-operational wildlife hunting block, which was moderately grazed (Moderate), and a hunting block, which was intensively grazed (Intensive). Boxplots indicate the mean (× within boxes), boxes range from the 25% to 75% quartile, and the end of the whiskers show the 5th and 95th percentiles. Different small letters denote statistically significant differences across treatments by Tukey's HSD test at *p* = 0.05.

### 3.2. Illegal Grazing Reduces Soil Penetration and Infiltration in Non-Operational Wildlife Hunting Blocks

Soil penetration capacity and soil infiltration rate differed significantly across different grazing categories ($F_{2,315}$ = 1295.7, *p* = 0.001 and $F_{2,315}$ = 441.8, *p* < 0.001, respectively), with the mean soil penetration capacity in moderate and intensive blocks being 46% lower than the control blocks (Figure 3A). A similar trend was also observed for the mean soil infiltration rate, 63% lower in moderate and intensive blocks than in the control blocks ($F_{2,315}$ = 441.8, *p* < 0.001; Figure 3B). Further, soil penetration and infiltration differed significantly across sampling period ($F_{2,315}$ = 3.0, *p* = 0.051 and $F_{2,315}$ = 4.9, *p* < 0.001). Soil penetration capacity in the hunting block of October 2019 period was slightly higher than in July 2019 and July 2020 ($F_{2,315}$ = 3.0, *p* = 0.051; Figure 3A) while soil infiltration rate across all treatments was 24% higher in October 2019 than in the other two sampling periods ($F_{2,315}$ = 4.9, *p* < 0.001; Figure 3B). The grazing categories × sampling periods showed significant interactions on soil penetration capacity ($F_{4,315}$ = 6.9, *p* < 0.001) and soil infiltration rate ($F_{4,315}$ = 4.9, *p* = 0.05; Figure 3). Soil infiltration rate was 11% and 17% lower in the moderate and intensive blocks, respectively, implying that grazing had a stronger effect on soil infiltration rate than the sampling period (Figure 3).

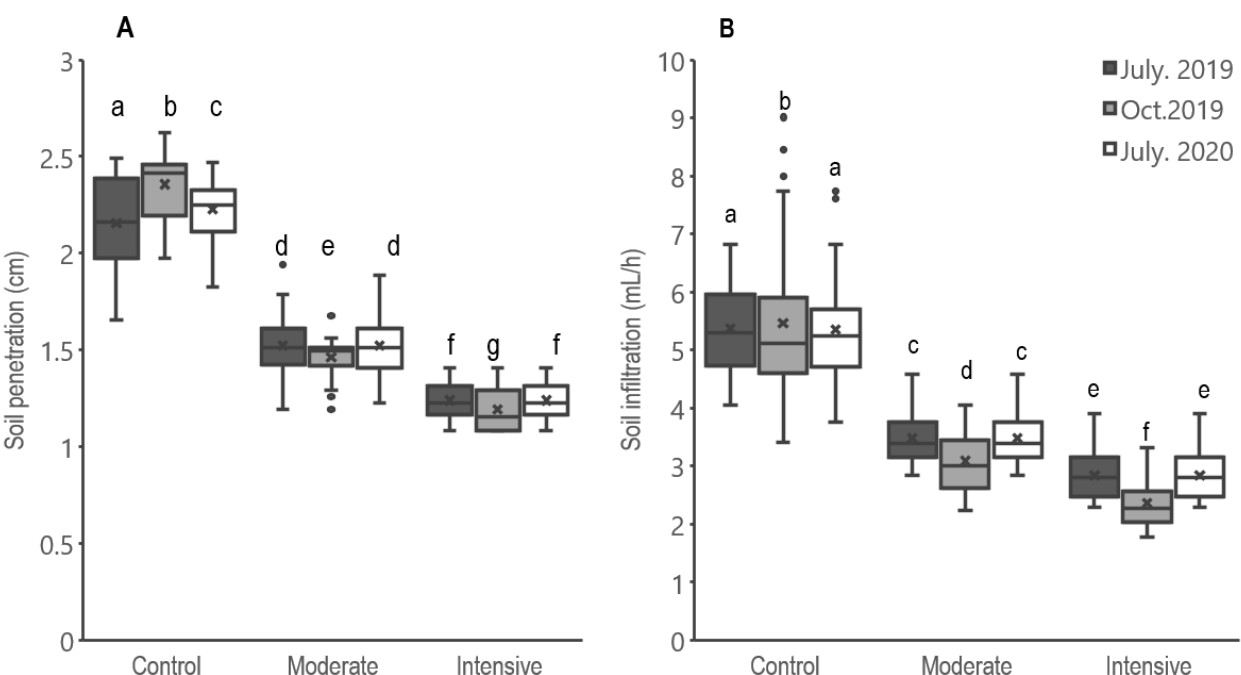

**Figure 3.** Boxplot of soil penetration capacity (**A**) and soil infiltration rate (**B**) across different grazing intensity and sampling periods in operational wildlife hunting blocks with little or no livestock grazing (Control), non-operational wildlife hunting blocks that were moderately grazed (Moderate), and hunting block that was intensively grazed (Intensive). Boxplots indicate the mean (the × within boxes), ranges from 25% to 75% quartile, and the whiskers' tips indicate the 5th and 95th percentiles. Different small letters denote statistically significant differences among treatments by Tukey's HSD test at *p* = 0.05.

### 3.3. Illegal Grazing Affects SOC and N-P-K in Non-Operational Wildlife Hunting Blocks

We found that soil organic carbon (SOC) was up to 28% lower in intensive and moderate blocks compared to the control block ($F_{2,33}$ = 8.0, *p* < 0.002) while it did not differ between intensive and moderate blocks ($F_{2,33}$ = 8.00, *p* = 0.964; Figure 4A). Contrary to our expectations, total nitrogen (TN) was 50% and 25% higher under intensive and moderate blocks than in the control block, respectively ($F_{2,33}$ = 32.2, *p* < 0.001; Figure 4B). Similar trends were seen for soil K *(H (2)* = 23.9, *p* < 0.001), which was 56% higher in intensively grazed block than in control blocks (Figure 4C). We found no significant difference in soil K between control and moderately grazed wildlife hunting blocks *(H (2)* = 23.87, *p* = 0.113; Figure 4C). Soil P remained similar across all hunting blocks *(H (2)* = 1.69, *p* = 0.428; Figure 4D).

### 3.4. Soil Penetration and Infiltration Rate Increases with Increasing Grass Biomass and Cover

Soil penetration and soil infiltration significantly increased with increasing grass cover (*R* = 0.61, *p* < 0.001; *R* = 0.52, *p* < 0.001, respectively). The same significant trend was visible for grass biomass and soil penetration (*R* = 0.80, *p* < 0.001) as well as soil infiltration (*R* = 0.75, *p* < 0.001; Figure 5).

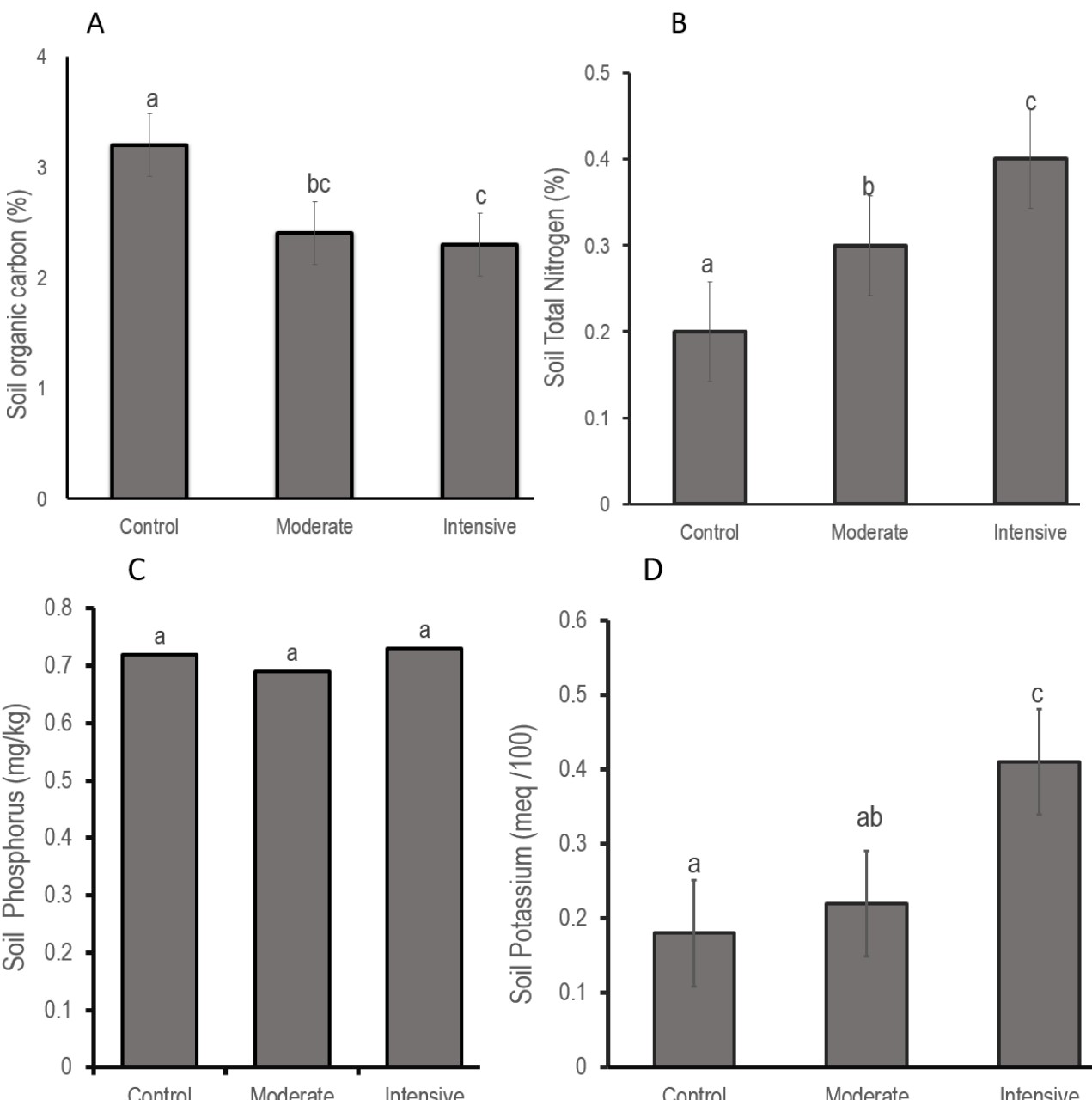

**Figure 4.** Mean (±SE) of available Soil Organic Carbon (**A**), Soil Total Nitrogen (**B**), Soil Phosphorus (**C**), and Soil Potassium (**D**) sampled from an operational wildlife hunting block without illegal livestock grazing (Control), non-operational blocks with moderate (Moderate), and intensive livestock grazing (Intensive). Different letters denote significant differences across hunting blocks with varying grazing intensities according to the post-hoc test at $p = 0.05$.

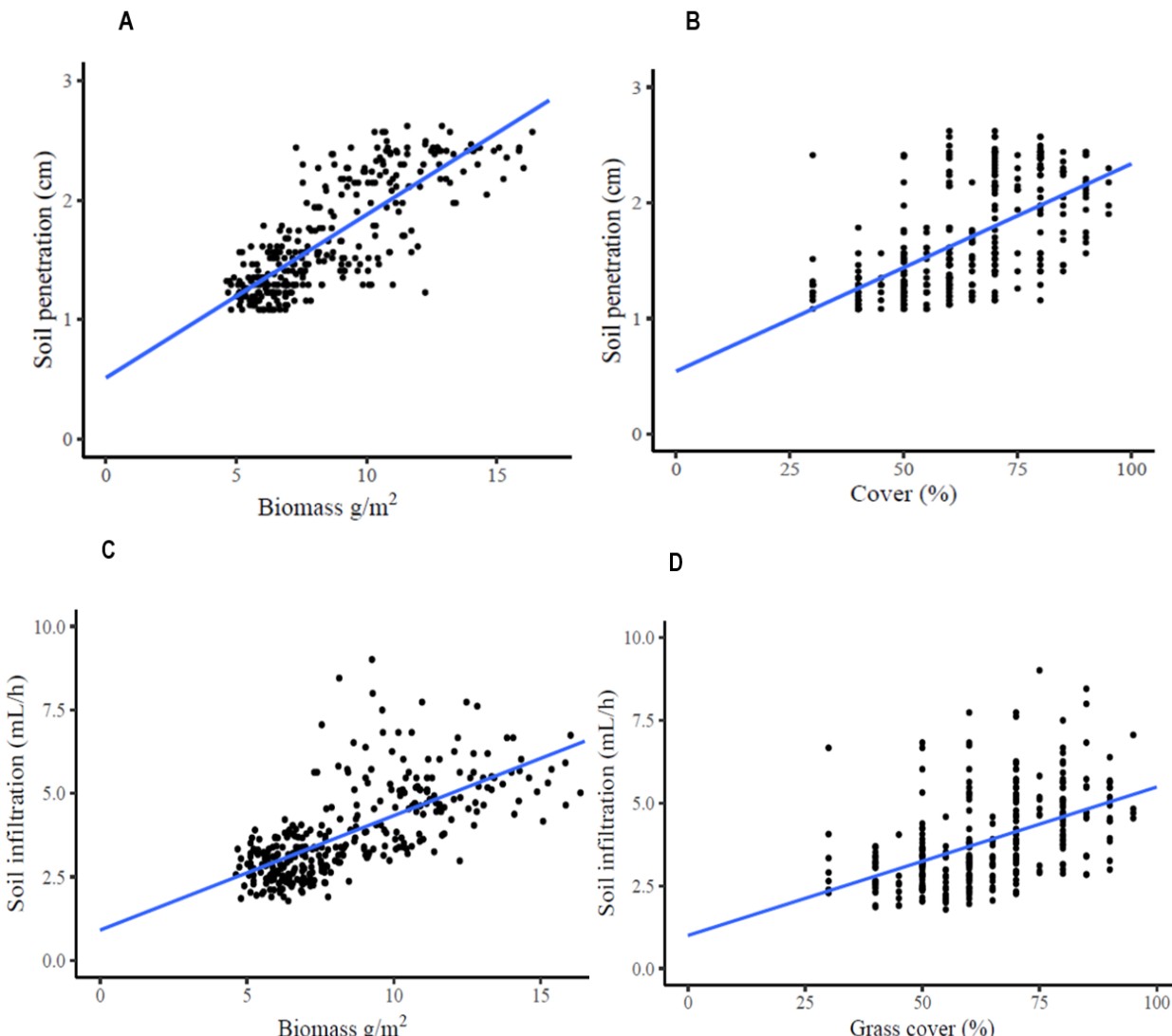

**Figure 5.** Correlation between (**A**) grass cover and soil penetration, (**B**) grass biomass and soil penetration, (**C**) grass cover and soil infiltration, and (**D**) grass biomass and soil infiltration in the Moyowosi-Kigosi Game Reserve. N = 324 collected across three sampling periods of July 2019, October 2019, and July 2020.

*3.5. Hunting Company Complements Anti-Poaching Efforts of the Game Reserve*

There is variation in patrol efforts exerted by HC and GR ($F_{1,14}$ = 18, $p < 0.001$). HC contributed an average of 347 worker-days$^{-1}$ for anti-poaching, which is 49% more than the anti-poaching efforts conducted by the GR for the same period (Figure 6A). Further, during the dry season, the HC's patrol efforts contributed 397 worker-days$^{-1}$, 63% more than the patrol efforts exerted by the GR (Figure 6B). There is also a significant difference between patrol success for GR and HC ($F_{1,21}$ = 116, $p < 0.001$), the mean patrol success for GR is higher than that of HC by 89% and 98% during dry and wet season, respectively (Figure 6B).

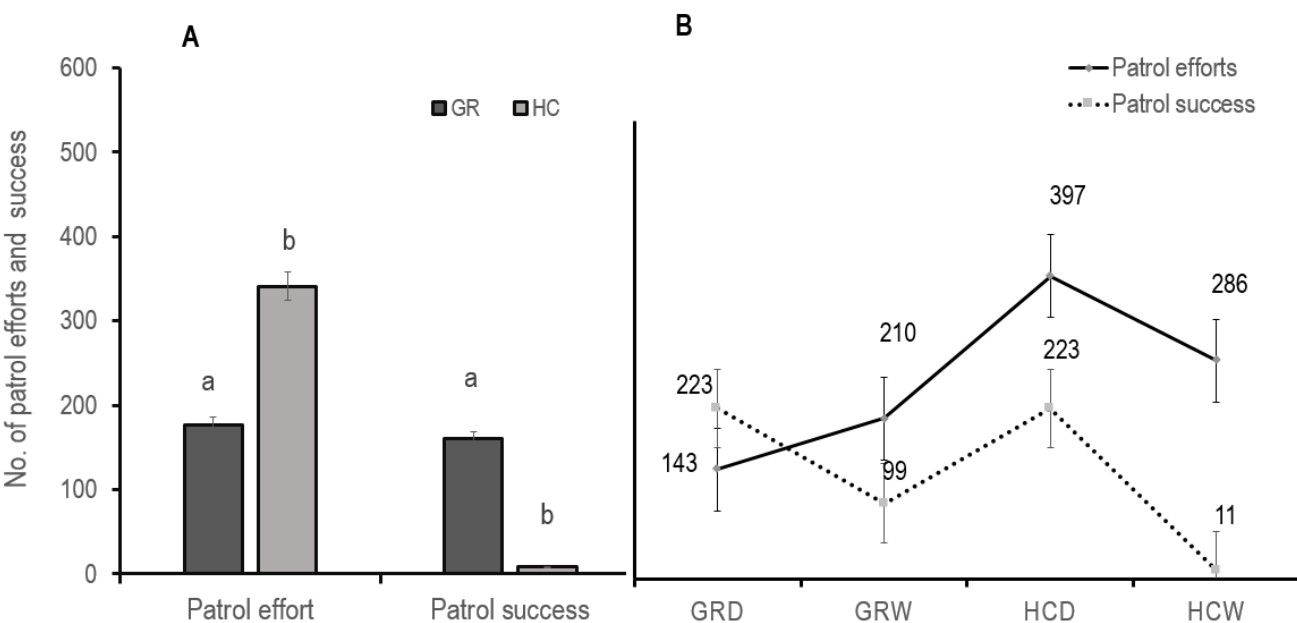

**Figure 6.** (**A**) is the mean (±SE) patrol efforts and patrol success of Game Reserves (GR) and Hunting Company (HC). Different letters denote significant differences in patrol efforts and success across GR and HC at $p$ = 0.05. (**B**) is the mean (±SE) patrol efforts and success by Game Reserve during the dry and wet season (GRD, GRW) and patrol efforts and success by Hunting Company during the dry and wet season (HCD, HCW), respectively. The solid line (—) represents patrol efforts, and the dashed line ( . . . .) represents patrol success. The numbers represent the mean patrol efforts and success.

## 4. Discussion

### 4.1. Low Grass Biomass and Cover in Non-Operational Wildlife Hunting Blocks

Our current findings show that illegal grazing negatively affected grass biomass and cover, as was supported by other researchers in temperate and semiarid grasslands [44,45]. The sampling period of October 2019 produced higher ABG and grass cover than other periods, highlighting the importance of rainfall for vegetation productivity. October 2019 was extraordinarily wet in the region compared to other sampling periods [46], which strongly stimulated grass growth in MKGR. However, when the grass layer was subjected to intense grazing by livestock, the effect of high rainfall was minimized, and grasses were reduced, which agrees with [47,48], who found grazing affected grassland production more strongly than rainfall. The high amount of rainfall further limited the rangers' access to the reserve, stimulating more illegal grazing without being detected by rangers and, thus, reduction in ABG and grass cover. However, our findings contradict studies by [49–51] in Africa and Asia, which found that rainfall increased AGB grass cover, irrespective of the grazing regime.

Our reduction of AGB and grass cover, despite the high amount of rainfall, could be caused by changes in soil structure (pugging) and loss of soil strength (poaching), caused by livestock trampling in wet soil [52,53]. The MKGR is a wetland reserve with black cotton soil [54,55], which strongly favors pugging and poaching [52]. Soil pugging and poaching can damage and even bury plants [56], hinder vegetation productivity [57] and increase the soil's resistance to return to its normal conditions [56]. Our findings that the negative effect triggered by grazing increased with wetness corresponds with previous studies by [58,59]. Thus, intensive livestock grazing in wet soil of MKGR affects highly important habitats resources that wildlife depends on for forages, reproduction, and shelter [58]; this might be irreversible and, thus, diminish populations of wild grazers in the long term in this fragile ecosystem [60].

*4.2. Reduced Soil Infiltration and Penetration in Non-Operational Wildlife Hunting Blocks*

As expected, we found that soil penetration capacity and soil infiltration rate were higher in operational wildlife hunting blocks (control) than non-operational wildlife hunting blocks. Soil infiltration rate and soil penetration capacity are indices for healthy soil [53,61]. The wetness of MKGR, dominated by black cotton soil, makes it more vulnerable to soil pugging and poaching [52]. Trampling of one hoof of a cattle can compact soil to an approximate depth of 5 cm, and the trampling pressure of one grown cattle weighing 350–600 kg is 200 kPa if standing and 400 kPa if it is moving [52]. This pressure reduces pore spaces of the soil, compacts soil [58], decreases soil infiltration rate and penetration capacity [62]. Our livestock density estimates in MKGR were 0.02 cattle ha$^{-1}$ in moderately grazed hunting block and 0.05–0.10 cattle ha$^{-1}$ in the intensively grazed hunting block. Our findings align with previous studies by [63,63–65] who found grazing increased soil compaction and reduced the ability of water to enter the soil of Mongolia and Nigeria.

*4.3. Mixed Results on SOC and N-P-K in Non-Operational Wildlife Hunting Blocks*

Our result of a lower SOC with higher grazing intensity could be caused by livestock biting, which removes premature grass leaves, damages photosynthetic tissues, decreases $CO_2$ fixation in plants [66], reduces leaf area for photosynthesis, affects grass growth [28], lowers AGB and cover and, thus, important soil organic material [67]. Our current grazing observations to reduce SOC are also in line with previous studies by [45,66], who found increased grazing reduced SOC in semiarid grassland China. The SOC is an important indicator of soil quality [66] as it improves soil structure, reduces bulky density, enhances the ability of soil to retain water [68], and boosts ecosystem functioning [66].

Contrary to our hypothesis, we found high soil TN and K in intensively and moderately grazed blocks.

Livestock uses 10–25% of nitrogen obtained from their pasture for their physiological metabolism and remove 75–90% from their body as waste, and add TN to the soil [15,62,69,70]. Livestock treading can further add N to the soil, as treading changes leaves, bark, and grasses that have fallen physically, increasing their decomposition rate and, thus, returning the amount of soil nitrogen to the soil [69]. Though soil nitrogen is essential for vegetation growth and productivity [70], excessive soil nitrogen can also increase invasive species and reduce variation of vegetation community, as was shown in the Mojave Desert [71]. High soil N also decreased flavonoids and phenolic characteristics of sesame seeds in Morocco [72]. Flavonoids protect plants from stresses, while phenolic gives plants structure and firmness [72]. Additionally, plants store excessive nitrogen in leaves as nitrates [70]; a high concentration of nitrate can negatively affect human health [73,74]. Our current observation of N to increase with increasing grazing intensity resembles previous studies by [69,75–77]. Similarly, livestock defecation increases soil K, as 85% of the ingested K is deposited back to the soil, primarily through urine [78]. Furthermore, ref. [15] also observed the increase of soil K following high grazing intensity in North-Eastern Iran. Livestock uses 20–30% of ingested P for their metabolism, and 70–80% is deposited back into the soil [78,79], which might explain our similar P values across grazing intensities. The defecated 70–80% P mixed into the soil produces fodders of high quality, which is eventually depleted by livestock grazing and create a balance between ingested and defecated P [79]. Moreover, livestock trampling disintegrates litter and phosphorus into the soil surface [80]. Unlike N, which is concentrated more in the livestock urine, a large amount of P is found in livestock dung [81]. Our findings of soil phosphorus stability, regardless of grazing intensity, correspond with [28], who found that as grazing intensity increased, the soil P remained stable in the Qinghai-Tibetan Plateau, China.

*4.4. Soil Penetration and Infiltration Rate Increase with Increasing Grass Cover and Grass Biomass*

As expected, we found a linear relationship between soil penetration capacity, soil infiltration rate, grass cover, and AGB, which agrees with the findings of [82]. Livestock trampling reduces ABG, grass cover, compact soil and makes it vulnerable to surface

runoff, evapotranspiration, and wind erosion [75,76,81–84]. When AGB and grass cover are well-maintained, the grass layer protects soil, increases organic matter, improves soil structure, and enhances soil infiltration and penetration capacity [15,16].

*4.5. Hunting Company Complements Patrol Efforts in the Game Reserve*

HC complements patrol efforts by the physical presence of hunting activities and by employing village game scout to conduct anti-poaching activities. The patrol effort exerted by HC during the dry season is higher than that of the wet season because hunting activities are conducted during the dry season (July to December). In the dry season, HC increases their patrol efforts to ensure maximum protection of their quality trophies and their clients (tourist). When trophy hunters, professional hunters, and game trackers search for quality trophies, they add eyes and boots to that particular hunting block [85,86]. Due to constant surveillance in their hunting blocks during the wet and dry season, the encountered number of unlawful activities is low. The fact that illegal activities decreased in hunting blocks after constant surveillance by the HC could imply a contribution of HC towards anti-poaching success [2,35]. During the rainy season, poaching in Eastern Selous increased following the closing of hunting activities [87].

On the other hand, despite the low patrol efforts exerted by GR during both dry and wet seasons, they encountered many illegal activities than HC. This could be explained by three factors; first, during the dry season, most of the MKGR is accessible, rangers' patrol unit can cover vast areas for surveillance and therefore encounter a high number of illegal activities; second, in the wet season, most parts of MKGR are inaccessible, which attracts high influx of poachers and illicit herders who uses this as an opportunity and invades in MKGR because they know they will not be detected easily. Even though there are limited patrol efforts during this season, rangers apprehend high number of illicit people in areas that can be accessed or through informers due to their high influx in wet season. Lastly, the absence of hunting activities in the surrendered hunting blocks attracts more illegal acts [88]. Owing to insufficient resources to support rangers perform their patrol effectively, especially during the wet season, the presence of the hunting company strongly supported the anti-poaching effort. It is possible to have MKGR with minimal or free illegal grazing if its accessibility is improved and patrol facilities supplied. This will enhance constant surveillance during the dry and wet seasons. Therefore, we recommend the MKGR construct watershed roads and deploy facilities such as an amphibious boat to improve anti-poaching, increase surveillance in the vacant hunting blocks, and attract more investors.

## 5. Conclusions

Overall, our findings show that illegal grazing strongly impacts wildlife habitat quality by decreasing AGB, grass cover, soil infiltration rate, and soil penetration capacity. The vacated wildlife hunting blocks were most affected due to insufficient anti-poaching efforts, which might also reduce resident buffalo populations due to intense grazing competition with livestock. Therefore, we conclude that trophy hunting contributes directly to wildlife habitat preservation by deploying various anti-poaching measures, including constant surveillance, which prevents illegal livestock grazing. Thus, we claim that it is vital to maintain trophy hunting as a key socio-ecological factor in conserving wildlife Protected Areas.

**Author Contributions:** Conceptualization, N.V.M. and J.V.W.; Formal analysis, N.V.M. and A.C.T.; Investigation, N.V.M.; Methodology, N.V.M., A.C.T. and P.A.N.; Supervision, J.V.W., P.A.N. and A.C.T.; Writing—original draft, N.V.M.; Writing—review & editing, J.V.W., P.A.N. and A.C.T. All authors have read and agreed to the published version of the manuscript.

**Funding:** This research did not receive any specific grant from funding agencies in the public, commercial, or not-for-profit sectors.

**Institutional Review Board Statement:** Not applicable.

**Informed Consent Statement:** Informed consent was obtained from all subjects involved in the study.

**Data Availability Statement:** Not applicable.

**Acknowledgments:** We are grateful for the support provided by the Management of Moyowosi-Kigosi Game Reserve to provide all important data and a ranger during our data collection in the field. We would also like to thank the Local authority who gave us permission to conduct interviews in their respective Districts and the field work team for their cooperation during field work design and data collection.

**Conflicts of Interest:** The authors declare no conflict of interest.

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
