# Peer review of "Using Trophy Hunting to Save Wildlife Foraging Resources: A Case Study from Moyowosi-Kigosi Game Reserves, Tanzania"

_sustainability, doi:10.3390/su14031288_

Round 1

Reviewer 1 Report

In general, this is an interesting manuscript that seems to deserve being published in the journal sustainability. There is a tendency of over-interpretation, my estimation is that 1/3 of the discussion could be shortened without losing relevant info.

However, one aspect became more and more obvious when reading: there are too many careless and avoidable formatting mistakes reducing the joy of reading. This is especially pronounced in the abstract but also in the main text, Figure 5 exists twice, there are too many spaces inserted between words, the spelling of terms (capitalized vs. small letters) is not consistent. One might argue that these are small formatting issues but to me, it shows that the authors should spend more time with the text and be more serious with such issues. The manuscript needs fine-tuning…  In the following I refer to passages of the text:

Abstract:

Line 26 and 27: Hunting Company (HC) is spelled out and abbreviated twice, after first mentioning in the abstract, only the abbreviation may be used.

Line 28: why is “GR” placed in parentheses here?

Line 29: why is there a closing-bracket behind the term “GR” but no opening bracket anywhere?

Line 30: instead of “F (1,21)” the authors should spell: “F(1.21)”

Line 32: Almost at the end of the abstract, the authors first mention the name of the study area, which is also abbreviated, although this abbreviation does not appear again in the abstract: “Moyowosi-Kigosi Game Reserve (MKGR)”, my recommendation would be to mention the area somewhere at the beginning of the abstract

Introduction:

Line 77: although in use, I`d recommend to avoid the term “Bos taurus” for the taurine cattle and rather stick to a more precise nomenclature scheme viewing the taurine cattle as the taurine form of the extinct aurox: “Bos primigenius f. taurus” see for example:  https://doi.org/10.1016/j.gecco.2019.e00756

Lines 76-80: despite the effect of livestock on wild herbivorous mammalian species in terms of competition for nutritional resources, disturbing effects of livestock, and cattle in particular, may have several reasons, see for example: https://doi.org/10.1016/j.gecco.2020.e01124

Line 82: the authors here spell “protected areas” in small letters, while in line 63, they use capilatized letters: “Protected Areas”  - please be concise…

Lines 101-105: This is info on methods and should be part of the following chapter, it could be the beginning of the material and methods chapter

Material & methods

Line 113: The phrase “- the world’s second deepest freshwater lake globally” contains a redundancy.

Lines 116-125: please be precise with the spelling of English names of species (capitalized, e.g. “Giraffe” vs. non-capitalized, e.g. “blood wood” or “white rhino” [line 57….], “bighorn sheep [line 53]), my recommendation would be to only capitalize proper nouns, such as “African buffalo” or “African elephant” but “giraffe” or “shoebill”….

Line 131-132: the info, whereas “…the aerial census of 2014 showed that livestock numbers inside MKGR were more than 50,000 heads” would gain relevance if authors could compare it with a livestock number from a previous year or the number of a certain species inside the area, e.g. estimated number of African buffalos or so….

Line 138: caption: “hunting blocks” not “hunting block”…

Line 191: Please remove the term “the” in the phrase: “while we the used Semi Micro Kjeldahl method for total nitrogen

Results:

Line 214: The heading “Illegal grazing lowers grass biomass and covers in non-operational wildlife hunting blocks” is misleading, one could assume the term “covers” is a verb: do the authors mean: “cover”?

Line 275: Please spell the letter “p” small but not capital.

Figure 5 c and d: For the sake of a better comparability, I recommend to use the same scale for the y-axes of the two graphs (soil infiltration between 0.0 and 10.0ml/h)

Line 298: “p” but not “P”, see comment above

Line 281 vs. 296: both Figures are termed “Figure 5”, the second Figure 5 should be Figure 6 and this should also be adapted in the text.

Line 287: I don`t understand the value / unit “347 person-days -1”. From the figure, I can see that there were 347 patrols – but why person per days? I also do not find appropriate explanation in the methods section – please clarify

Discussion:

In view of the results presented, the discussion appears a bit lengthy, and the manuscript would probably benefit from a shortening of the discussion without necessarily loosing info.

Line 313: please check the use of parentheses here – is the literature appropriately referenced? The same applies to line 319 and other – carefully check the guidelines of the journal.

Line 316-319: It appears that the authors present a result that was not presented in the results section? I strongly recommend to stick to the scientific structure: intro, methods, results, discussion…

Line 313: check spaces

Line 319: check spaces

Author Response

Dear Reviewer,

We are grateful for your valuable inputs which has enabled us to improve our paper. Please find the attached addressed comments

Kind regards on behalf of all authors

Nyangabo Violet Musika

Reviewer 2 Report

Needs clarification of what is meant with success - is it lack of poaching? is it encountering fewer poachers or more? Needs to be made very clear in the methods!

a few issues with referencing - sometimes commas are used other times they are missing in the brackets.

Line 48 ' Despite the critics' should that be 'despite the criticism'?

Line 52 'donated' are these donated or are these fees? or both - not clear.

Line 56 'was conserved' ...small populations 'were conserved' plural?

Line 57 'hunting business promoted'  - revenue doesn't promote - it can assist with the promotion of something as it allows the purchase of certain services etc?

There seem to be a few formatting issues with extra spaces appearing in various places?

Line 96 N-P-K it appears that the acronym has not been explained before using it here for the 1st time

Line 173 Escali Primo model - might need a reference?! There are other methods / models etc within this MS that have not been referenced - some are common but for an international audience it might be good to supply some refs to the original sources?

Line 427 'opportunity ad invades' not sure what that 'ad' means?

Ref list might have a couple of formatting issues!

Author Response

Dear Reviewer,

Thank you very much for your valuable contributions which has improved or paper.Please find attached response that address your comments.

Best regards on behalf of all authors

Nyangabo Violet Musika
